# Exosomes in Liquid Biopsy: The Nanometric World in the Pursuit of Precision Oncology

**DOI:** 10.3390/cancers13092147

**Published:** 2021-04-29

**Authors:** Karmele Valencia, Luis M. Montuenga

**Affiliations:** 1Program in Solid Tumors, Center for Applied Medical Research (CIMA), 31008 Pamplona, Spain; 2Consorcio de Investigación Biomédica en Red de Cáncer (CIBERONC), 28029 Madrid, Spain; 3Navarra Health Research Institute (IDISNA), 31008 Pamplona, Spain; 4Department of Biochemistry and Genetics, School of Sciences, University of Navarra, 31009 Pamplona, Spain; 5Department of Pathology, Anatomy and Physiology, School of Medicine, University of Navarra, 31009 Pamplona, Spain

**Keywords:** exosomes, cancer, liquid biopsy, biomarkers

## Abstract

**Simple Summary:**

Exosomes are small vesicles of 100 nm in size that are released from every cell constantly. They contain different molecules (DNA, RNA, lipids, metabolites, etc.) that reflect the content of the cell they come from. Exosomes can be found in all biological fluids. In cancer, exosomes are involved in several events such as tumor growth, metastasis, and the immune response, by delivering their cargos to recipient cells. Due to their unique features, exosomes have become promising analytes in the field of liquid biopsy, which searches for biomarkers to manage different steps of the tumor process. We believe that exosomes will become an important tool in liquid biopsy in the near future. In this review we provide an updated literature compilation about exosomes as biomarkers in oncology and discuss their possibilities and limitations.

**Abstract:**

Among the different components that can be analyzed in liquid biopsy, the utility of exosomes is particularly promising because of their presence in all biological fluids and their potential for multicomponent analyses. Exosomes are extracellular vesicles with an average size of ~100 nm in diameter with an endosomal origin. All eukaryotic cells release exosomes as part of their active physiology. In an oncologic patient, up to 10% of all the circulating exosomes are estimated to be tumor-derived exosomes. Exosome content mirrors the features of its cell of origin in terms of DNA, RNA, lipids, metabolites, and cytosolic/cell-surface proteins. Due to their multifactorial content, exosomes constitute a unique tool to capture the complexity and enormous heterogeneity of cancer in a longitudinal manner. Due to molecular features such as high nucleic acid concentrations and elevated coverage of genomic driver gene sequences, exosomes will probably become the “gold standard” liquid biopsy analyte in the near future.

## 1. Exosome Biogenesis and Composition—Reflecting Their Origin 

Exosomes are extracellular vesicles (EVs) with a size range of ~40 to 160 nm (average ~100 nm) in diameter with an endosomal origin. All eukaryotic (and also prokaryotic) cells release exosomes as part of their active physiology [1].

Exosomes are generated in a process of sequential invagination of the plasma membrane that results in the formation of multivesicular bodies (MVBs), which can intersect with the trans-Golgi network, endoplasmic reticulum, or other intracellular vesicles, contributing to the content heterogeneity of exosomes. Within the cell, the MVB can either fuse with lysosomes or autophagosomes to be degraded or fuse with the plasma membrane to release the contained vesicles (exosomes). Exosome biogenesis is reflected in the presence of a variety of proteins either integrated in their membrane or as exosomal cargo: small Rab family GTPases; annexins and flotillin; Alix, Tsg101, and ESCRT complex; tetraspanins CD9, CD63, and CD81; or heat shock proteins Hsp70 [2,3,4,5]. ExoCarta, an exosome database (http://exocarta.org/; accessed on 27 April 2021), has been developed to identify exosomal contents. Approximately 10,000 different proteins have been characterized in relation to the exosomal component [6]. Figure 1 shows the main exosome components.

How DNA is contained in exosomes is far from being resolved and is still controversial. It has been shown that DNA-containing micronuclei that originate from nuclear membrane collapse can interact with exosomal tetraspanins, leading to the shuttling of the DNA in MVBs [7]. Also, mitochondria produce vesicles containing mtDNA that reach the endolysosomal system to form MVBs (reviewed in [8,9]).

Exosome production varies depending on the cellular origin, metabolic status, and cellular microenvironment. One unresolved question about exosomes today is to distinguish tumoral-origin exosomes from non-tumoral counterparts. Moreover, it is still unclear how exactly the exosomal content is selected and loaded into vesicles and how exosomal trafficking is regulated. To solve these questions, it is crucial to fully understand the biology of exosomes. This better knowledge is an essential requirement for future clinical applications of exosomes as diagnostic (and even treatment) tools.

## 2. Exosomes: A Source of Biomarkers 

The path towards more precise and personalized management of cancer patients is currently focused on the development of novel non-invasive biopsy technologies that are easy to obtain, may be repeated over time to follow longitudinally the progression of the disease, and may be able to reflect the phenotypic and genetic heterogeneity of the tumor. Liquid biopsy (LB) offers all of these potential benefits. LB is based on the search for biomarkers that may help clinical decision making. Those biomarkers may be applied to screening/early diagnosis, prognosis, prediction of response or resistance to treatments, detection of minimal residual disease, confirmation of relapse, disease monitoring, etc.

Among the different components that can be analyzed in liquid biopsy, the utility of EVs is particularly promising because of their presence in all biological fluids and their potential for multicomponent analyses. The concentration of analytes in membrane-surrounded vesicles may potentially allow for higher sensitivity and specificity over other types of liquid biopsy looking for single and even multiplexed free circulating biomarkers [10,11]. Exosomes are the most abundant analyte within the liquid biopsy, reaching 1 × 10^11^ particles per milliliter of blood. In an oncologic patient, up to 10% of all the circulating exosomes will be tumor-derived exosomes depending on tumor stage [12]. Figure 2 describes liquid biopsy analytes and their concentrations. Exosome content mirrors the features of its cell of origin in terms of DNA, RNA, lipids, metabolites, and cytosolic/cell-surface proteins. In addition, exosome content has a number of advantages in comparison to other liquid biopsy analytes. First, exosomes contain high-quality RNA that can be extracted from fresh or frozen fluids. Second, different types of RNA are contained in exosomes, including miRNA [13,14], piwi-interacting RNA, pseudo-genes, lncRNA, tRNA, and mRNA including different splice isoforms found in the cells of origin. Third, exosomes are released from viable tumor cells. Furthermore, their DNA recapitulates the entire genome and the mutational burden of the parental tumor, a great advantage compared to ctDNA, where DNA is fragmented. Evidently, it is significantly more difficult to obtain information about the specific DNA alterations pursued in a given analysis or, worse, to obtain the sequence of the entire genome from highly fragmented circulating DNA [15,16]. In addition, as exosomes contain both RNA and DNA (reflecting tumor mutations), the use of a single platform to study both molecular species is a clear advantage for finding rare or not-abundant mutations. Finally, the protein content of a single exosome reaches up to 400 unique proteins [17,18].

Therefore, the fact that exosomes include several molecules that can be considered as potential biomarkers, alone or in combination, increases the possibility of success in the pursuit of a good LB biomarker, which is a clear advantage over the other LB analytes. Moreover, the number of released exosomes could also be considered a clinical indicator itself (see Section 3).

Table 1 summarizes information rendered by different analytes of LB and shows the potential clinical applications of them as biomarkers.

## 3. Exosome Heterogeneity: An Unknown Wealth?

Exosomes constitute a heterogeneous population of vesicles. This heterogeneity arises from the combination of different parameters such as cellular origin, content, size, number, and functionality. These parameters interact directly with each other, making it very difficult to isolate one without entering the field of another. Within an organ, exosomes can be released from epithelial (tumoral or normal) cells, as well as from stromal cells, lymphocytes, etc. This discrimination could be possible due to the preservation of cell-type-specific membrane proteins on the exosome membrane. There have already been reports in the literature of some examples where, using well-known specific proteins found in exosomes, researchers were able to easily recognize and differentiate exosomes with breast or pancreatic origin [20,21].

The cellular origin of exosomes will determine their composition, at both the membrane and soluble levels. Therefore, the second factor that creates heterogeneity among exosomes is their content. The content of exosomes also varies in response to many factors. It responds to different cellular stages such us metabolic wellness [22]. Thus, an exosome’s hallmarks will dynamically change as a result of the modifications that occur in their cell of origin. Moreover, tumor-derived exosomes (TEX) expressing different integrins or other molecules in their membrane have been related to different organotropisms similar to what is shown in tumor spreading cells [23,24] and, more interestingly, TEX are uptaken with greater affinity by certain cell types within an organ [25].

The content of an exosome is limited by its size. This brings us to the third parameter of heterogeneity. Exosomes are also a mixed population in terms of size. As previously mentioned, exosomes show a size range of ~40 to 160 nm. Therefore, a 150 nm diameter exosome will be able to contain a greater number of molecules than a smaller exosome. It is still unknown whether different exosome sizes respond to distinct cellular stages or cause diverse responses in target cells, but what have been reported in recent studies are significant differences in the number and size of exosomes in cancer patients depending on the studied biological fluid [26].

The number of exosomes released from a cell is another source of heterogeneity. Due to the constant influx of exosomes, the exosomal release–uptake dynamics of different cells, and the lack of fine characterization of exosome origin, it is difficult to ascertain whether the amount of TEX is different compared to that from normal cells. Historically, it has been demonstrated in vitro that tumor cells secrete more exosomes than their normal cell counterparts. Thus, different studies reported higher exosome protein amounts in cancer patients than in healthy controls [27] (reviewed in [10,28]). However, technological studies in breast cancer pointed to the opposite situation, where the capture of shed exosomes in a single-cell platform showed lower numbers of exosomes in tumor cells compared with tissue-matched, nontumorigenic cell-line-derived exosomes [29]. Such studies relied on different isolation methods, experimental designs, and quantification methods, facts that can easily disturb results. Therefore, further investigation is needed to clarify this important aspect of exosome biogenesis. The literature describes an increased number of total circulating exosomes in the peripheral blood of cancer patients and, surprisingly, their size and morphology are also altered compared to those of healthy donors [30]. More interestingly, recent studies showed significant differences in the number and size of exosomes in cancer patients depending on the studied biological fluid [26]. This fact highlights the importance of selecting an ideal bodily fluid as a tool for the search and study of exosome-based biomarkers in each given type of cancer. In summary, the underlying mechanism of these alterations during the tumor course is unclear. 

The final source of heterogeneity we will refer to is exosome functionality. Exosomes show very diverse effects on the cells that uptake them. The consequences are so varied that we dedicate an epigraph below to exploring the most studied and characteristic outcomes (Figure 3).

Taken together, these data suggest that exosome heterogeneity might play a dual role in the characterization of a patient’s tumor. On the one hand, the number and other above-mentioned hallmarks of exosomes could give us a clue about the tumoral stage and its possible progression, but on the other hand, this mix could dilute valuable information in their use as accurate biomarkers. Exosomal-related biomarkers are discussed below.

## 4. Sending a Message: The Role of Exosomes in Intercellular Communication

Exosomes have been shown to provide a natural mechanism for cell-to-cell communication, with a plethora of roles in physiology and pathology. In every communication process, a relationship between a sender and a receiver is established through the emission and reception of a “message” that will have an impact on the recipient. Exosomes are known to play a very important role in the communication process between tumor cells and their microenvironment. Recently, several groups visualized through elegant imaging techniques the process of exosome uptake in NSCLC [31] and breast cancer [32]. The content of tumor-released exosomes can be uptaken by other adjoining tumor cells, tumor-niche (stroma) cells, immune cells, or distal organ cells after travelling through the circulatory system.

There are still many questions about the role of exosomes in intercellular communication. For example, it is still unknown how different outcomes on receptor cells may be affected by uptake affinity differences between recipient cell types or by different modes of exosomal uptake (receptor-mediated endocytosis, direct binding, direct fusion, etc.) [33]. The regulation of the different potential cellular fates of the cargo transported by the uptaken exosomes is also not clearly known. The contents of the exosomes can be directly transferred to the degradation pathway or may be secreted into the endoplasmic reticulum and/or to the cytoplasm. Specific membrane transport mechanisms may be involved in these different inner cellular outcomes. Furthermore, it is plausible that depending on the nature of the exosomal cargo and the state of the recipient cell, the ability of the exosomal message to affect specific recipient cell functions may be variable, which makes the understanding and the study of the exosomal-based communication process even more complex.

As just mentioned above, exosomes have an impact on the recipient cells that will influence the development of the tumoral process. Exosomes have been described to be involved in different neoplastic stages such as tumor growth, metastasis, and resistance to therapy, contributing to different hallmark features of cancer (Figure 4) [34]. Many of these hallmarks will appear in the following section.

In order to better explain the role of exosomes in intercellular communication and the effects that they trigger in recipient cells, we divided this epigraph into two sections, using the distance to which the receptor cell is located as the criterion. We focus on TEX examples as in the cancer field this is the central and most studied population of exosomes.

### 4.1. A Short-Range Shipment: The Role of Exosomes in the Tumor Microenvironment

Epithelial-to-mesenchymal transition (EMT) is a process through which cancer cells may become more proliferative and resistant and gain migratory and invasive properties [35]. TEX are thought to be partially responsible for this cellular plasticity, inducing EMT in adjoining tumor recipient cells [36] through the modulation of several well-known signaling pathways. Thus, regulation of Wnt/β–catenin or PI3K/AKT in human lung cancer cell lines [37,38] and modulation of the Hippo and ERK pathways [39,40] in hepatocellular carcinoma upon TEX uptake have been reported. Similarly, activation of AKT signaling triggered by TEX has been reported to induce EMT [41].

Classically, the field of exosomes has focused on understanding how TEX uptake by stromal cells modifies the tumor niche, modulating the microenvironment to favor tumor development. Most studies have focused on defining the functional changes in cancer-associated fibroblasts (CAFs) and immune cells [42]. 

In this sense, several systems of TEX-mediated immune suppression have been described. TEX carry ligands that bind to cognate receptors on immune cells, inducing tolerogenic signaling [43] and inhibiting tumor-specific T cells [44]. The response of activated T cells to TEX interaction triggers a reduction in both JAK expression and the response to IL-2 [45,46] which prevents them from proliferating. Furthermore, TEX carry CD39 and CD73, which activate the adenosine pathway, a well-known immunosuppressive factor that inhibits T-cell function [47,48]. More interestingly, TEX carrying FasL [49] or programmed death ligand 1 (PD-L1) induce the apoptosis of activated CD8+T cells by triggering both extrinsic and intrinsic apoptosis pathways [50]. Importantly, FAsL and PD-L1 exosomal expression levels correlate to spontaneous apoptosis of circulating T cells and to tumor prognosis [51]. Recently, it was reported that the suppression of exosomal PD-L1 induces systemic anti-tumor immunity and memory [52]. On the contrary, TEX lead the differentiation and expansion of Tregs [44,53]. TEX also modulate NK cytotoxicity by downregulating NKG2D expression, which suppresses NK cell activity [54]. Besides this, tumor-derived exosomes inhibit monocyte differentiation into DC cells [55], directly inhibiting DC bioactivity and inducing immune tolerance [56]. However, TEX skew monocyte differentiation into myeloid-derived suppressor cells (MDSCs) [57,58], which accumulate in murine tumor, spleen, peripheral blood, and lung in vivo [59]. This fact negatively affects antigen processing and presentation and produces several immunosuppressive inhibitory factors, including NO and ROS, causing TCR nitration or T-cell apoptosis [60]. Moreover, neutrophils that uptake TEX DNA increase IL-8 and tissue factor production, boosting tumor inflammation and paraneoplastic events (thrombosis) [61]. TEX also generate an immunosuppressive microenvironment by activating macrophages to a tumor-associated macrophage (TAM)-like phenotype [62,63]. Finally, the role of exosomes in the innate immune response has also been described in cancer. TEX were shown to harbor B cells and exert a decoy function limiting complement-mediated lysis and decreasing cytotoxicity against cancer cells [64].

TEX are also implicated in angiogenic remodeling, an essential step in tumor survival, growth, and dissemination, through favoring new vessel formation [25] or destroying the integrity of the endothelium and promoting vascular permeability and metastasis [65].

Tumor dissemination is also facilitated by TEX triggering matrix destruction by MMP1 activation [66].

The reciprocal exchange of exosomes between tumor cells and CAF has also been a focus of study. In this way, CAF-derived exosomes support the metabolic fitness of cancer cells growing as tumors through several known mechanisms, such as switching mitochondrial oxidative phosphorylation to glycolysis [67] or promoting motility via Wnt-planar cell polarity (PCP) signaling [68].

### 4.2. A Long-Range Shipment: The Role of Exosomes in Metastatic Organs 

Primary tumor TEX can reach metastatic organs through the circulation (blood or lymphatic). It has been described how the pattern of integrins present in the exosome membrane determines TEX organotropism. Thus, breast cancer TEX bearing integrins α6β4 and α6β1 were associated with lung metastasis, while exosomal integrin αvβ5 was linked to liver metastasis [23]. Once exosomes have reached the metastatic organ, they are uptaken by specific cells in those organs, and the message they carry is translated by the receptor cells into microenvironment remodeling orders known as premetastatic niche preparation [69], an essential change for the nesting and engraftment of circulating tumor cells (CTCs) reaching the metastatic organ. The recruitment of bone marrow progenitor cells and macrophages to metastatic sites is one of the changes related to TEX that are involved in premetastatic niche formation and enhance metastatic potential [70,71]. Also, TEX prevent patrolling Ly6C low monocyte expansion, enabling immunosuppression and leading to metastasis [72]. The activation of cancer-associated fibroblasts (CAFs) is also involved in premetastatic niche formation. TEX can trigger TGF-b signaling pathways and thereafter initiate a program of differentiation of fibroblasts toward a myofibroblastic phenotype, altering the stroma which will be then responsible for supporting tumor growth, vascularization, and metastasis [73]. In turn, CAF-derived exosomes induce oxidative phosphorylation in metastatic breast cancer cells, contributing to their exit from the dormant state [74]. Figure 5 recapitulates the role of TEX locally and in distant organs. 

## 5. TEX Biomarkers in Clinics: A List of Possibilities 

One of the great proposals in the field is to study exosome contents as potential biomarkers. Despite biological fluids being composed of a complex mixture of molecules (RNA, DNA, and proteins), diagnostic approaches have traditionally focused on a single molecular species. In the case of exosomal cargo, the same trend has happened. RNA is the most abundant and studied exosomal component, being unusually stable thanks to its exosomal membrane confinement. In 2012, the National Institute of Health (NIH) dedicated a strategic Common Fund to the study of exosomal RNA (http://commonfund.nih.gov/Exrna/index; accessed on 27 April 2021). Since then, the interest in the field has continuously increased. The number of entries related to exosomes, RNA, and cancer in PubMed has increased more than 10-fold from 2012 to date. 

Currently, the focus of translational studies is also turning to exosomal DNA assessment, with more than 200 publications being found in PubMed in 2020. Probably, one of the most specific hallmarks of cancer is DNA mutations, which can also be captured within exosomes [75]. Many recent works take advantage of existing technologies for circulating free DNA (ctDNA) detection in LB. The translational use of exosome DNA sequencing is an exciting approach that still needs to be fully explored and developed. 

Proteins contained in exosomes also include altered proteins associated with cancer. In addition, exosomal surface proteins are related to the functional status of the cells comprising the tumor immune microenvironment, which may be important biomarkers for monitoring response to immunotherapies [52].

Due to their multifactorial content, exosomes constitute a unique tool to capture the complexity and enormous heterogeneity of cancer [10]. To bring exosome-based liquid biopsy diagnosis closer to the clinic, several high-throughput platforms have recently been developed. Among them, microfluidic devices based on antibody-capturing systems in microchips [76,77,78,79,80] seem to be the best option for clinical application [81]. These novel technical approaches aim to make exosome-based diagnostics cost and labor effective, by means of developing highly sensitive and reproducible detection devices to isolate and identify circulating cancer markers without using a large volume of sample and sparing the time-consuming ultracentrifuge-based isolation processes that are usually associated with exosome analysis. 

Multicomponent diagnostic/prognostic applications based on exosomes are currently being considered. These high-throughput multiplexed analyses can also be combined with deep-learning-based interpretation methodologies, which will require pilot studies in large numbers of clinical samples [82]. These approaches may overcome the sensitivity and specificity of current biomarkers in a number of clinical situations. Moreover, the inclusion of exosome cargo analysis in the biomarker laboratory armamentarium may help to characterize not only the tumor but also its microenvironment, leading to a more accurate tumor description and understanding.

The table below (Table 2) summarizes a list of exosomal analytes proposed as biomarkers in the last three years. All of them have been studied in well-characterized cohorts of patients. Nevertheless, in a considerable proportion of these examples, especially in the case of miRNAs, validation in independent cohorts, together with robust statistical criteria and harmonized protocols, is still needed. The main disadvantage of working with exosomes is still the lack of technical consensus, which leads to poor inter-laboratory reproducibility of the results. Therefore, before exosome-based biomarkers become a clinical reality, major efforts have to be made to standardize every single procedure in exosomal-based biomarker studies: isolation, characterization, and analytical protocols [83].

## 6. Future Perspectives and Challenges: The Dawn of a New Era

Liquid biopsy applications have been exponentially growing since 2010. According to RNCOS market research, the global liquid biopsy market is expected to reach 5 billion dollars by 2023 [103]. Among the different analytes in LB, circulating tumor DNA (ctDNA) seems to be the one with the most promising results in the field. The main bottleneck of ctDNA-based LB is to develop technologies sensitive enough to measure low amounts of ctDNA in circulation, particularly when early detection or minimal residual disease is pursued. Next-generation sequencing (NGS)-based technologies have reached a compromise between sensitivity and cost and they are already available in clinical laboratories. By August 2020, the FDA had approved the first two blood tests, Guardant360 CDx and FoundationOne Liquid CDx, as companion diagnostic tests that provide molecular information (mainly specific mutations or CNA) predictive for the effective use of associated drugs in NSCLC, prostate, breast, and ovary and for general tumor profiling in solid tumors [104,105]. Previous NGS-based tests approved for use in DNA extracted from FFPE or other tumor tissue samples have previously shown great efficacy as companion biomarkers.

Although many efforts have been made in detecting ctDNA in blood, it is worth mentioning that ctDNA seems to be mainly released passively from dying normal or tumor cells (necrosis or the different types of programmed cell death). It is also actively shed from neutrophils by the process called NETosis [106]. However, DNA can also be released within exosomes in an active and selective manner. In fact, it has been reported that more than 93% of amplifiable cfDNA in blood is in fact found as cargo of plasma exosomes [107]. Therefore, exosomes are potentially very valuable raw materials for more sensitive analysis of circulating DNA, as DNA is highly concentrated in exosomes released from tumor and other cells. To date, only a few studies have compared the clinical parameters of “gold standard” ctDNA and exosomal DNA (exoDNA). Only in pancreatic ductal adenocarcinoma, KRAS mutation detection in exoDNA was superior to ctDNA for prognosis [98,99]. It has also been shown that the combination of exoDNA/RNA and ctDNA has better sensitivity and specificity than ctDNA alone for EGFR T790M mutation detection in NSCLC [93,94] and BRAF V600E mutation detection in melanoma [97]. 

Despite the need for more studies in large patient cohorts to evaluate exoDNA as a circulating biomarker, preliminary data are very promising. High exosomal nucleic acid concentrations and elevated coverage of the genomic driver gene sequences will probably help to make the analysis of exosomes the “gold standard” LB DNA-based analyte in the near future. 

In summary, although the field of exosomes in liquid biopsy is still immature, its potential for the very near future seems enormous, promising, and fascinating. The major hurdles for exosomal-based biomarkers to reach the clinic are the standardization and optimization of isolation and characterization methodologies and the validation of reported results in multiple independent cohorts. 

In fact, there are many different techniques to isolate exosomes. They can be classified into five main groups according to the chemical or physical isolation system: centrifugation, precipitation, affinity binding, microfluidics, and molecular size-exclusion-based techniques. Each method has its pros and cons. In general, an exosome isolation technique with elevated yield numbers will render low exosome purity, and vice versa. Therefore, the isolation method may be adapted to respond to each specific need. It is important to take into consideration such factors as the type and amount of initial sample or the subsequent use of those isolated exosomes. Moreover, some other aspects will determine the final choice of the technique, e.g., the need for specialized equipment, cost, time, or scalability. Table 3 summarizes the pros and cons of the main exosome isolation technologies.

Although there is still no consensus on a standard isolation method, the International Society for Extracellular Vesicles (ISEV) is making an strong effort to achieve this aim [83].

Also, understanding of the regulatory mechanisms that control tumor-derived exosome heterogeneity that may influence the reproducibility of diagnostic outcomes is essential. 

In addition, the development of liquid-biopsy-based multiparametric assays is expected to return large data sets of different nature (nucleic acids, proteins, etc.). For this reason, the implementation of artificial intelligence tools for data management and analysis, as well as the development of models that include all complex exosome-derived data, is starting to be explored to accurately use exosomes as cancer biomarkers [108].

In summary, the research avenues for the near future in the field of exosomes in cancer liquid biopsy are multiple, wide, and very exciting.

## Figures and Tables

**Figure 1 cancers-13-02147-f001:**
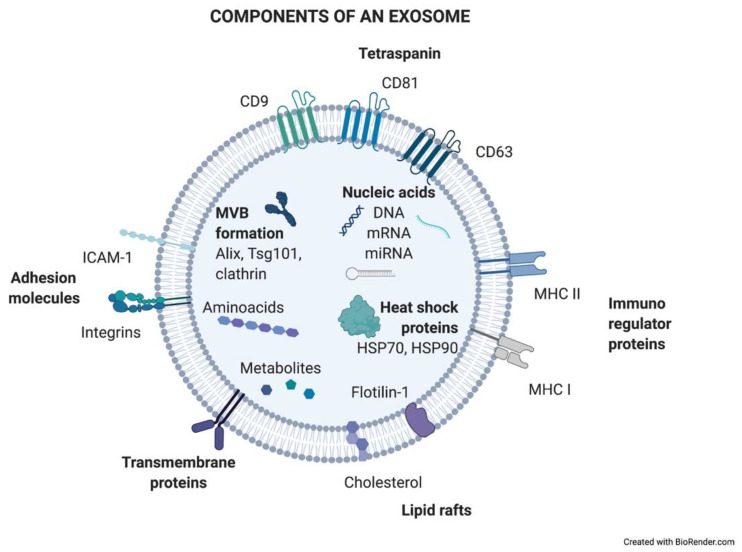
Components of an exosome. Exosomes contain a wide variety of molecules of different natures, such as nucleic acids, proteins, or lipids. All the content at both the membrane and soluble levels represents the cell of origin the exosome is release from.

**Figure 2 cancers-13-02147-f002:**
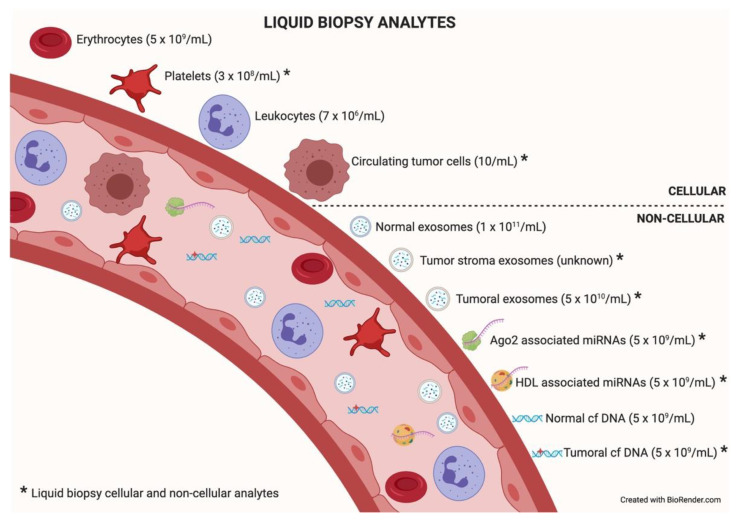
Liquid biopsy analytes. In the bloodstream, many components can be found, cellular or non-cellular in nature. Some of them constitute liquid biopsy analytes (marked with an asterisk). Higher concentrations of analytes (in parentheses) will facilitate isolation techniques and subsequent analysis. Data taken from [12].

**Figure 3 cancers-13-02147-f003:**
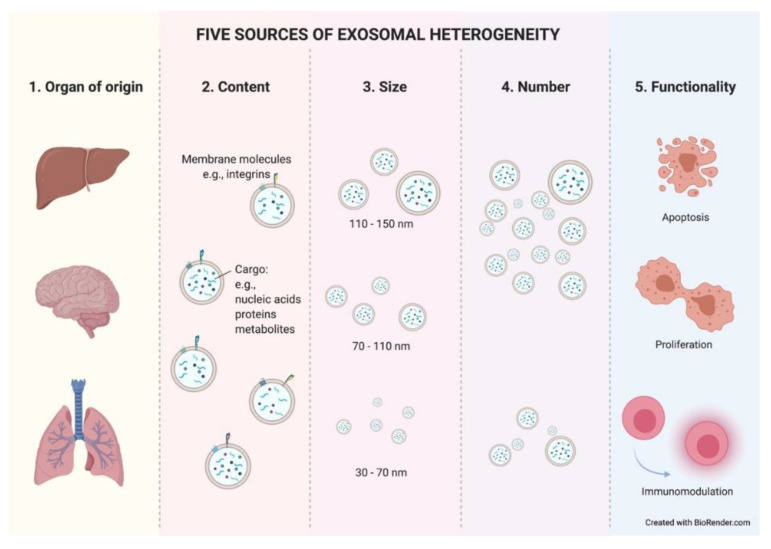
Five sources of exosomal heterogeneity. The heterogeneity of the exosomes results from the combination of five factors: the cell of origin from which they are released (organ and cell type of origin); their molecular composition; their size; their number; and the functionality triggered in recipient cells. Different combinations of these five factors make exosome heterogeneity highly complex.

**Figure 4 cancers-13-02147-f004:**
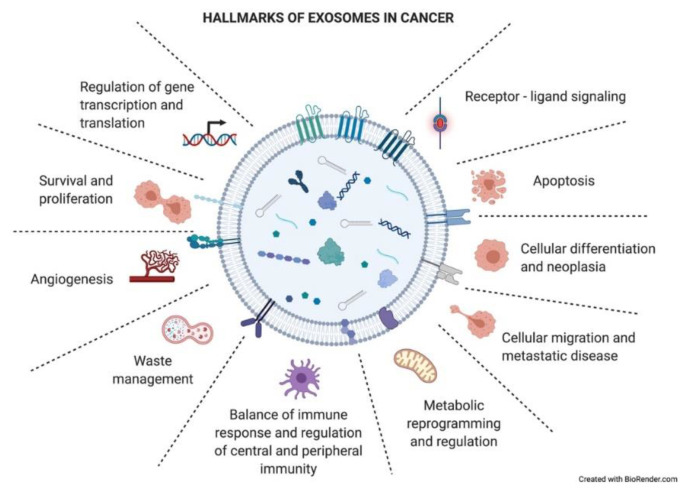
Hallmarks of exosomes in cancer. Tumor-derived exosomes have important functional roles in intercellular crosstalk, affecting the biology of their target cells in different manners. Through this crosstalk, exosomes drive the tumoral process and other pathological conditions. The picture summarizes responses that exosome uptake can trigger in the recipient cells (functional hallmarks).

**Figure 5 cancers-13-02147-f005:**
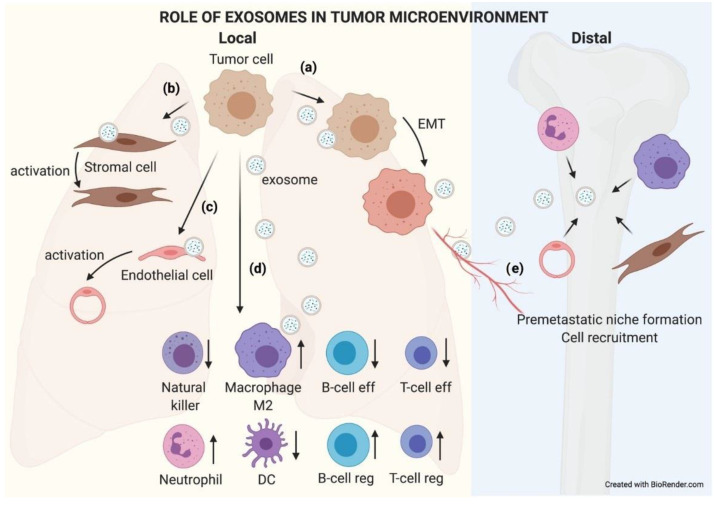
The role of exosomes in the tumor microenvironment. TEX communicate with their microenvironment. At a local level, exosomes can be uptaken by other adjoining tumor cells, favoring an EMT process (**a**). They may also modify the biology of stromal (**b**) and endothelial (**c**) cells by activating them to support the tumor. Exosomes received by immune cells favor an immunosuppressive microenvironment that helps tumor growth (**d**). Furthermore, exosomes can reach blood or lymphatic vessels and travel to distal organs, where they will educate native cells, preparing a premetastatic niche where subsequent circulating tumor cells (CTCs) will nest and grow (**e**).

**Table 1 cancers-13-02147-t001:** Liquid biopsy analytes: features, extractable information, and clinical applications as biomarkers. Table adapted from [19].

Traits	Liquid Biopsy Analyte				
	CTCs ^1^	ctDNA ^2^	Exosomes	ctRNA ^3^	miRNA
Origin					
Viable cells	✔ ^4^	✖ ^5^	✔	? ^6^	?
Apoptotic cells	✔	✔	?	?	?
Components					
DNA	✔	✔	✔	N.A. ^7^	N.A.
RNA	✔	N.A.	✔	✔	✔
Proteins	✔	N.A.	✔	N.A.	N.A.
Metabolites	✔	N.A.	?	N.A.	N.A.
Extractable information					
Copy number variation	✔	✔	✔	✖	✖
Mutations	✔	✔	✔	✔	✖
Epigenetic information	✔	✔	✔	✖	✖
Fusion genes	✔	✔	✔	✔	✖
Splice variants	✔	✖	✔	✔	✖
Single-cell information	✔	✖	✖	✖	✖
Application in personalized medicine					
Diagnosis	✔	✔ ^8^	✔	?	✔
Classification of molecular subtypes	✔	✔	?	?	✖
Clonal evolution tracking	✔	✔	?	✖	✖
Prognosis	✔	✔	✔	?	✔
Recurrence	✔	✔	✔	✔	✖
Predictive	✔	✔	✔	?	✖
Resistance prediction	✔	✔	✔	?	✖
Monitoring treatment	✔	✔	✔	?	?

^1^ Circulating tumor cell; ^2^ circulating tumor DNA; ^3^ circulating tumor RNA; ^4^ yes; ^5^ no; ^6^ no data; ^7^ not applicable; ^8^ most probably.

**Table 2 cancers-13-02147-t002:** Examples of exosomal-derived potential biomarkers with clinical significance published in the last three years.

**Exosomal miRNAs as Cancer Biomarkers**				
miRNA	Cancer type	Clinical value	Biofluid	Reference
Let-7b-5p, -122-5p, -146b-5p, -210-3p, -215-5p	Breast cancer	Diagnosis	Plasma	[84]
miR-224	Hepatocellular carcinoma	Diagnosis/Prognosis	Serum	[85]
miR-106b, miR-1269a	Lung cancer	Diagnosis/Prognosis	Serum	[86,87]
miR-375, -1307	Ovarian cancer	Diagnosis	Serum	[88]
**Exosomal lncRNAs as Cancer Biomarkers**				
lncRNA	Cancer type	Clinical value	Biofluid	Reference
PCAT-1, UBC1 and SNHG16	Bladder cancer	Diagnosis/Prognosis	Urine	[89]
MALAT-1	Lung cancer	Diagnosis	Serum	[90]
**Exosomal mRNA as Cancer Biomarkers**				
mRNA	Cancer type	Clinical value	Biofluid	Reference
BRAF, KRAS (mutant)	Colorectal cancer	Diagnosis	Serum	[91]
**Exosomal mutated DNA as Cancer Biomarkers**				
DNA	Cancer type	Clinical value	Biofluid	Reference
IDH1	Glioblastoma	Diagnosis/Prognosis	Plasma	[92]
EGFR	Lung cancer	Diagnosis/Prognosis	Plasma/Bronchioalveolar lavage	[93,94,95,96]
BRAF	Melanoma	Therapeutic monitoring	Plasma	[97]
KRAS, P53	Pancreatic cancer	Diagnosis/Prognosis	Serum/Plasma	[98,99]
MYC, P53, MLH1, PTEN, AR	Prostate cancer	Diagnosis/Prognosis	Plasma	[100,101]
**Exosomal proteins as Cancer Biomarkers**				
Protein	Cancer type	Clinical value	Biofluid	Reference
PDL-1	Melanoma	Prognosis	Plasma	[102]

**Table 3 cancers-13-02147-t003:** Pros and cons of the main exosome isolation techniques.

Factors	Ultracentrifugation		Precipitation	Affinity		Microfluidic	Filtration	
	Differential	Gradient		Immune	Flow Cytometry		Ultrafiltration	Molecular Exclusion
Purity	low	high	low	high	high	high	low	high
Yield	medium	low	medium	medium	medium	low	medium	high
Specialized equipment	medium	medium	high	medium	low	low	high	high
Specialized user	medium	low	high	medium	medium	medium	high	high
RNA characterization	high	high	high	high	high	high	high	high
Protein characterization	medium	high	low	high	high	high	medium	high
Functional studies	medium	medium	low	medium	medium	medium	medium	high
Scalability	medium	low	high	medium	high	low	medium	medium
Time	medium	low	high	medium	low	medium	high	medium
Cost	high	medium	high	low	low	low	medium	medium

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
