# Peer review of "Exosomes in Liquid Biopsy: The Nanometric World in the Pursuit of Precision Oncology"

_cancers, 2021, doi:10.3390/cancers13092147_

Round 1

Reviewer 1 Report

The review by Valencia and Montuenga examines the use of exosomes as liquid biopsy and propose them as promising and powerful diagnostic tool in cancer.  The outline of the manuscript  is acceptable and well described, but I suggest to make some corrections as reported below, and to include and discuss some specific points, that will complete the overview.

  1. Figure 1, the word reachs, may be the authors mean reaches??? Regarding the feature of a good biomarker in cancer “stable over time” can authors discuss  in the text  on the stability of exosomes? Was this issue  treated in other studies, or it  was treated only in terms of pharmacokinetics and biodistribution of exosomes (see doi.org/10.1016/j.xphs.2017.02.030)? In case add some sentences on this topic in the text.

  1. Line 76, correct having and integrated with having an integrated

  1. Table 1 . In the figure legend the symbol X is missing. What it is for??

  1. Actually,  the protein content of a single exosome is much higher than 100 molecules, since given certain assumptions of protein size and configuration, and packing parameters, it can reach about 20,000 molecules (Maguire G,:1016/B978-0-323-41533-0.00007-6). Please modify this point.

  1. Line 125, rephrase the title

  1. Line 153, change the word content with soluble

  1. Line 155. I suggest to introduce a sentence describing the well known exosome specific proteins reflecting the cell from which they originate , as for example Mart-1  for melanoma derived exosomes (Mears R, 2004 ),  epidermal receptor (HER)  in exosome from breast cancer, and pancreatic cancer origin, (Ciravolo V, Journal of cellular physiology. 2012; Adamczyk KA,. Life sciences. 2011), that could render tumor exosomes easily distinguishable from exosome secreted by non-tumor cells.

  1. Line 191-194. This conclusion is not obvious. The authors state that the exosome heterogeneity is an added value in  the characterization of a specific tumor. Whereas this is true in terms of exosome amount that generally increases with the disease progression, and heterogeneous  biological functions  as reported in Figure 3, hovewer it can represent an obstacle in the field of exosomes as an elected LB biomarker. In fact,  can the absence  of uniformity in a given population guarantee or provide the essential requirements  for a precise diagnosis? Please explain this point in the text.

  1. The authors briefly mention in the text the absence of optimized protocols of exosome isolation that render the field of exosomes in LB still immature. However the lack of an elective procedure to isolate pure exosome population within the bulk of EVs highly secreted by tumor cells is more than an issue, and I believe that it deserves a discussion in the text. Therefore, I suggest to make examples of the techniques used for LB with a brief explanation of advantage/

Author Response

We thank this reviewer for his/her kind suggestions and comments.

Reviewer 2 Report

In the review article, the authors summarized the aspects of exosomes in oncology. While I believe the topic is of interest to audience of Cancers and exosome holds great potential in oncology, the article is not organized to deliver a clear landscape of exosome. It is packed with superficial claims and random examples, with a lack of focus and critical discussion.  A major revision is required in order to publish on Cancers of high quality.

  1. The article needs to start with an introduction part to give an overview of exosomes.

  1. Among the 10 sections, the first 2 sections are all about liquid biopsy. Reader won’t know it is about exosome until section 3. Authors should consider moving Section 4 to beginning and condense the biospy part.

  1. Similarly, Fig 1, Fig2 and Table 1 are all about biospy and biomarkers. Need to be condensed. Authors mentioned that Table 1 and Fig2 are adapted from Ref. 6 and 10, but I don’t see the relevant data in the 2 papers. Please clarify.

  1. Among the 10 sections, sections 6, 7 and 8 are similar, they are all about the role of exosome in signaling/communication. Authors gave a lot of examples in these sections, but there is no clear logic, giving sections 7/8 low readability.

  1. Cargos delivered by exosome, such as protein and DNA, are an important aspect in liquid biospy. While authors mentioned these scatteredly, I would expect a separate section to discuss it. Tumor-derived exosome, which is very important and is discussed randomly in many places in the article, can form a separate section as well.

  1. Please reorganize section 5 by summarizing 4 listed sources of exosome heterogeneity first, and then explain 1 by 1 in each paragraph.  

  1. In Fig4 of section 6, authors listed ~10 hallmarks of exosomes in cancer, but didn’t discuss any one of them. Please at least talk about the most important ones.

  1. In summary, authors mentioned that a major hurdle for use of exosome-based biomarker in clinic is about isolation and characterization (L415). Please talk about the isolation and characterization methods of exosomes. This is important for any nanometric components.

  1. Authors said application of artificial intelligence in exosome should be explored to address unmet needs (L420). Please be specific as to what AI can do in exosome study and to solve what issues.

  1. Please make sure acronyms are defined at their first appearance only. TEX are defined a dozen times. There is no need to define common terms, such as ‘communication’ and ‘biomarker’

Author Response

We thank this reviewer for his/her suggestions and comments.

Reviewer 3 Report

Corrections:

figure 1:             wrong spelling reaches

line 76:               an instead of and

table 1:               missing space  ..tumor RNA, N.A.

line 103:             depending on tumor stage recommended

line 106:             exosome-content has a number of  advantages in comparison to other liquid biopsy

figure 2:             wrong spelling of erythrocytes, platelets, and leukocytes

line 154-155:     PSMA bad example, PSMA expression is found in extraprostatic tissues, including small bowel and brain, often published under different synonym, see https://www.prospecbio.com/psma_human I don´t see the relevant data excluding exosomes from the brain in the quoted literature (29) DOI: 10.3892/ijo.2014.2256, exchange and add a relevant clear-cut example

line 231:             exosome uptake

line 235:             add a citation on EMT e.g. https://doi.org/10.1016/j.tranon.2020.100773

line 310:             immunosuppression

line 310/314:     different separations which are correct based on separation rules e.g. met-a-stat-ic, but these frequent separations disturbs flow of reading throughout the review… another example lines 49, 55, 60… 329, 367, 391: bi-omarkers.

line 421:             exosomes

The paper would benefit by adding, how the isolation of exosomes of interest can be done e.g. ultracentrifugation, chemical precipitation and affinity-binding including discussion of the respective shortcomings.

Author Response

We thank this reviewer for his/her comments and suggestions.

Round 2

Reviewer 2 Report

The authors have significantly improved the structure of the review article and resolved all major issues raised in the first round. I recommend acceptance in the present form.